# Rehabilitation Capacity in South Africa—A Situational Analysis

**DOI:** 10.3390/ijerph20043579

**Published:** 2023-02-17

**Authors:** Quinette A. Louw, Thandi Conradie, Nolubeko Xuma-Soyizwapi, Megan Davis-Ferguson, Janine White, Marie Stols, Andronica Masipa, Pringle Mhlabane, Lungisile Mdaka, Claudina Manzini, Ivy Kekana, Marike Schutte, Simon Rabothata, Pauline Kleinitz

**Affiliations:** 1Division of Physiotherapy, Department of Health and Rehabilitation Sciences, Faculty of Medicine and Health Sciences, Stellenbosch University, Cape Town 7500, Western Cape, South Africa; 2Rehabilitation Services, Department of Health, Bhisho 5200, Eastern Cape, South Africa; 3Disabilities and Rehabilitation, Western Cape Department of Health, Cape Town 8000, Western Cape, South Africa; 4Western Cape Rehabilitation Centre, Western Cape Department of Health, Cape Town 7789, Western Cape, South Africa; 5Therapeutic and Rehabilitation Services, Limpopo Department of Health, Polokwane 0700, Limpopo, South Africa; 6Rehabilitation and Disability Services, Mpumalanga Department of Health, Nelspruit 1200, Mpumalanga, South Africa; 7Rehabilitation Service, Gauteng Department of Health, Pretoria 2001, Gauteng, South Africa; 8Sensory Functions, Disability and Rehabilitation Unit, Department for Noncommunicable Diseases, World Health Organization, 1211 Geneva 27, Switzerland

**Keywords:** capacity, public health sector, rehabilitation, South Africa, workforce

## Abstract

Rehabilitation in South Africa (SA) operates independently of major health services and reforms, despite the increasing rehabilitation need. With the introduction of National Health Insurance (NHI), SA is facing another major health reform. Evidence is needed on the current SA rehabilitation situation, regarding shortcomings, opportunities, and priority strategic strengthening actions. We aimed to describe the current rehabilitation capacity in the SA public health sector, which serves the majority and most vulnerable South Africans. A cross-sectional survey was conducted in five provinces, using the World Health Organisation’s Template for Rehabilitation Information Collection (TRIC). Participants were purposively selected for their insights and experiences of rehabilitation in specific government departments, health sectors, organisations, and/or services. TRIC responses were analysed descriptively. Participants explained how timely and effective rehabilitation produced long-term health, social, and economic benefits. Positive initiatives were reported for rehabilitation data collection, service design, and innovation. Challenges included inadequacies in human resources, the integration of rehabilitation at primary care, guidelines, and specialised long-term care facilities. The continuity of care across levels of care was sub-optimal due to inefficient referral systems. Promoting and improving rehabilitation nationally requires concerted, innovative, collaborative, and integrated efforts from multiple stakeholders within, and outside, the health system.

## 1. Introduction

*Rehabilitation:* Rehabilitation is an essential health service that improves the lives of people with a wide range of health conditions—including many people with disabilities—through optimising their day-to-day functioning. Rehabilitation outcomes may include improved mobility, personal care capacity, the ability to communicate, greater independence, as well as joining (or re-joining) the workforce or attending school or university. Rehabilitation is relevant for people of all ages—for instance, premature infants with developmental delay; children with neurological disorders, hearing, or vision loss; or people of any age recovering from illness, trauma, communicable diseases (e.g., tuberculosis [TB], HIV, COVID-19), or dealing with the sequelae of chronic diseases (e.g., stroke, arthritis, diabetes, heart problems, mental health disorders). People with compromised capacity are often unable to study or work, and hence cannot contribute optimally to their families, communities, or the country. Therefore, rehabilitation, more than the medical management of impairments, incorporates a holistic, person-centred approach to enable optimal reintegration (including social and economic) of people with disabilities and impaired functioning into their communities [1].

*Overview of South Africa’s health challenges:* There are many reasons why health status is compromised for many South Africans, and why so many cannot access the care they need. Firstly, the quadruple burden of disease, coupled with less-than-desirable social determinants of health [2,3] consistently ensures that many people cannot escape the never-ending cycle of poor health, disability, unemployment, lack of education, and poverty [4,5,6]. Secondly, non-communicable chronic diseases (NCDs) are rampant and largely unmanaged, with life-changing events such as stroke, diabetes, and heart disease affecting many [7,8]. Thirdly, the national burden of chronic diseases has increased dramatically since prevalent diseases like HIV (classified as an acute communicable disease five years ago), have recently been reclassified as chronic [9,10,11]. Fourth, the already huge national burden of acute and chronic disease is inflated by consistently high rates of trauma due to violent crimes, motor vehicle accidents, and workplace injuries for casual, unskilled, and uninsured workers [8,12,13]. Fifth, South Africa’s health profile is insidiously compromised by ongoing inequalities based on race, gender, socio-economic status, geographical area (e.g., rural/peri-urban/urban)—remnants of the Apartheid regime—and thus ongoing divides in the quality of (and access to) education and health resources for different population groups. Sixth, post-Apartheid governance continues to experience poor management, particularly affecting major portfolios of health, employment, and education [14]. There appears to be no imminent resolution for this situation. Finally, the alarming inequity between private and public sector healthcare continues, where those who can afford private care can access some of the best care in the world; while those who cannot, may wait weeks to access basic, and often inadequate, care. The public sector receives just under 50% of the total national healthcare funding (which includes state funding, health insurance, and out-of-pocket payments), but it serves the needs of over 80% of the population who rely on it for all their needs [15]. These constraints all challenge innovation and improvement in South African health services delivery.

*South Africa’s health reforms:* Since the first South African democratic election in 1997 (which marked the end of the Apartheid era), the government has embarked on variably successful health reforms, such as primary care re-engineering, to improve access to, and quality of, public healthcare for all South Africans [16,17]. However, rehabilitation has not been included in major reforms to date, despite the increasing need for rehabilitation due to the epidemiological transitioning from acute infectious diseases to chronic diseases and NCDs [17,18]. South Africa is on the cusp of another major health transformation that has the potential to redress many inequities—namely the National Health Insurance (NHI) [19], which is intended to provide a health safety-net for the most vulnerable in society. It is thus critical and timely to generate evidence on the current rehabilitation situation, to better describe and understand the shortcomings, opportunities, and priority strategic actions required for rehabilitation to become sufficiently integrated and capacitated within the envisaged NHI. Evidence of national rehabilitation capacity can inform, and reform, policies according to the growing population need for rehabilitation [18,20].

*Rehabilitation management:* In South Africa, policy and regulatory governance of rehabilitation falls within the remit of the National Department of Health (NDoH), but policy implementation and operational management is decentralised to provincial levels. A National Rehabilitation Policy was published in 2000, with strategic goals such as improving access to rehabilitation, appropriate resource allocation, human resource development, and the better integration of rehabilitation and rehabilitation-specific monitoring and evaluation frameworks [21]. Published reports suggest that little progress has been made towards achieving any goal [22]. Moreover, this policy has not been updated in the past 22 years. To clarify actions to assist with implementing the National Rehabilitation Policy goals, a Framework and Strategy for Rehabilitation and Disability was developed in 2016 [18]. To date, there is limited reporting or information from the nine South African provinces about what progress has been made in the National Policy goals. Furthermore, a study conducted by the Human Science Research Council in 2013 suggests that the public sector is not providing effective, efficient, or equitable rehabilitation services [23]. Because so many South Africans rely solely on the public sector for care, it is crucial to understand the current state of rehabilitation at all levels (national, provincial, district, and community) in order to ensure appropriate actions to integrate rehabilitation within NHI [24].

*Vertical information dissemination:* There are nine provinces in South Africa, with very different geography, resources, urbanisation, economics, and social dynamics. Each province has at least one dedicated rehabilitation manager, situated in the provincial health management structure, usually at a sub-directorate level in a larger departmental cluster, for example, Non-Communicable Disease Units (NCDUs). Provincial rehabilitation managers’ mandates involve managing rehabilitation at sub-national level; thus, they are the conduits between policy and action. However, the power of rehabilitation managers to make a difference to rehabilitation capacity is significantly affected by broader systemic challenges, such as fiscal and workforce constraints [12,17]. Poor financial management, low levels of provincial and national economic growth over the past decade, and the cross-sector impact of the recent COVID-19 pandemic indicate that rehabilitation may slip further down the list of priorities by policy makers, unless there is committed, unified action to change this trajectory.

*Global rehabilitation recognition:* The lack of rehabilitation services, and access to rehabilitation in South Africa, is not dissimilar to that found in other low- and middle-income countries (LMICs). The World Health Organisation (WHO) launched the global Rehabilitation 2030 Initiative, which aims to promote rehabilitation as the key health strategy of the 21st century around the world [25]. This assists countries to strengthen rehabilitation through multipronged approaches, including an assessment process which culminates in strategic planning as well as rehabilitation-specific monitoring and evaluation frameworks [26]. This approach recognises the specific challenges faced by LMICs, which often have higher burdens of disease and higher needs for rehabilitation, but must prioritise limited resources across different levels and type of healthcare [19]. The WHO has developed a comprehensive health system assessment tool, the Systematic Assessment of Rehabilitation Situation (STARS), which assists countries to interrogate the state of local rehabilitation and facilitates the prioritisation of strategic actions to improve local rehabilitation services. The tool was specifically developed to support rehabilitation assessments in LMICs where rehabilitation is often under-resourced and under-recognised, and poorly integrated into health systems. STARS includes the Template for Rehabilitation Information Collection (TRIC), for the collection of comprehensive and standardised information of the rehabilitation capacity in countries [27]. This tool is anchored in all major pillars of the health system (governance, finance, workforce, information, assistive technology, and services) as defined by the WHO [20,28]. A TRIC assessment provides governments with much-needed evidence to inform priority actions that seek to integrate and strengthen rehabilitation within countries’ health systems.

*South African contexts:* The core health professionals involved in rehabilitation in South Africa compares to global models and include physiotherapists, occupational therapists, speech-language and hearing therapists, orthotists, prosthetists, and audiologists, supported by the rest of the healthcare team, patients, and their families, carers, and communities. Rehabilitation training at an undergraduate level of core rehabilitation professions is well-established in South African universities, but remains mostly profession-specific, with little interprofessional collaboration in teaching, learning, or research. This siloed approach continues after therapists enter the workplace in service, management, or advocacy capacities [17]. This can complicate engagement with policy makers, as they may hear the same story in different ways—from discipline-specific advocates—about how rehabilitation links to key performance indicators, and population health. Not surprisingly, given professional fragmentation in rehabilitation disciplines and lobby groups, constrained resources, and multiple priority health demands in South Africa [3], there has been limited investment by the government in building rehabilitation capacity. However, the rehabilitation needs of South Africans are growing, in parallel with an increasing burden of disease and a broad range of negative consequences on health, which can be ameliorated if rehabilitation is provided [8].

## 2. Materials and Methods

### 2.1. Aim

To describe current public sector rehabilitation capacity in South Africa’s public healthcare system using the WHO TRIC tool, which incorporates all the WHO health system building blocks.

### 2.2. Collaborating Partners

The collaborating partners were the WHO, NDoH, Clinton Health Access Initiative (CHAI), and Stellenbosch University (SU). A concept note, outlining the terms and conditions for the partnership, was agreed to by all collaborating bodies. The NDoH provided administrative support and assisted with the coordination of provincial Departments of Health (DoHs). The WHO country, regional, and headquarter offices provided technical assistance. CHAI provided support with national networks and contacts for the situation analysis. SU led the assessment, obtained ethical approval, collected data, and was responsible for writing and sharing the project report with all partners.

### 2.3. Ethics and Permission to Participate

Permission to conduct the research was obtained in writing from all participating provincial DoHs. Ethics approval was received from the lead researcher’s institution (N19/04/048). Information about the project, invitations to participate, and consent forms were emailed to potential participants. Written and verbal consent from each participant was obtained prior to data collection, and researchers clarified, for each consenting participant, how their anonymity would be protected.

### 2.4. Data Collection Instrument

We applied the WHO TRIC tool, to enable comprehensive and standardised assessment of public sector rehabilitation in South Africa [27]. The TRIC also provided the framework for data analysis. The TRIC collects quantitative and qualitative information (closed- and open-ended questions) on current rehabilitation capacity, as well as new or emerging initiatives or opportunities in terms of:Governance, regulation, and leadership;Financing;Human resources;Rehabilitation information and research;Infrastructure and medications;Assistive technology;Rehabilitation services;Emergency preparedness.

Appendix B provides a summary of the TRIC tool content that was applied to this study. We focused our assessment on the public health sector since the majority (about 84%) of the South African population utilises the public sector of health care [29], and thus excluded questions on non-state rehabilitation services. Additional data collected on human resources were reported on in another published manuscript [30].

### 2.5. Study Design

A mixed-methods study with an embedded design was used [31]. Qualitative interviews were secondary to the cross-sectional survey and were used to strengthen or explain findings from the close-ended questions of the TRIC tool.

### 2.6. Setting

The study was conducted in five of the nine provinces of South Africa, which were purposively selected to reflect the country’s diverse socio-economic environments (representative of rural and urban locations, the presence of tertiary institutions where rehabilitation programmes are offered, and gross domestic income). Due to ethical requirements, the provinces will remain unnamed in this report.

About 67% of the population lives in the selected provinces, which, respectively, hosts 11%, 26%, 10%, 8%, and 12% of the population (StatsSA 2021 report). Two of the provinces are mostly urban and the most affluent in the country. The remaining three provinces are predominantly rural and among the poorest in the country. One of the provinces is the most populous and contributes most (34%) to the country’s Gross Domestic Product (GDP). One urban province contributes 14% to the GDP and the predominantly rural provinces contribute about 8% each (https://www.statssa.gov.za/?p=12056, accessed on 3 November 2022).

In South Africa, similar to most LMICs, the core professionals classified as rehabilitation professionals (providing physical rehabilitation) include occupational therapists, audiologists, speech-language therapists, orthotists, prosthetists, physiotherapists, and mid-level workers [32]. The two urban provinces collectively have five university rehabilitation training departments, which offer degree programmes in physiotherapy, occupational therapy, speech-language and hearing therapy, and audiology. There is only one university among the three rural provinces, which offers a speech and language training programme.

### 2.7. Sampling and Participants

Sampling was done in two stages. First, the research team purposively identified key informants (managers, lead clinicians, district coordinators) in each participating province, who held comprehensive knowledge of local rehabilitation systems, service delivery, context, resources, capacity, priorities, agendas, barriers, and rehabilitation workforce. Participants were purposively selected based on existing contacts and networks by participating partners. Second, key informants were asked to invite other colleagues whom they believed could provide important information about rehabilitation in their province. When no new informants were identified, sampling ceased. We believed that a heterogenous, representative pool of informants would enable us to build a comprehensive description of the broad range of factors related to rehabilitation services in South Africa and develop an understanding of provincial and site variability in rehabilitation resources, contexts, and service provision. We envisaged that the participants would include occupational therapists, speech therapists, physiotherapists, audiologists, orthotists, and prosthetists; since South Africa does not have physiatrists while the training of community rehabilitation workers in South Africa remains limited resulting in their contribution to rehabilitation being fragmented and not well implemented across the country [33]. In addition, 220 participants participated in the data validation sessions.

### 2.8. Data Collection

Between March 2020 and September 2021, the first author conducted three sessions with each participant (or group of participants, as appropriate) to assist them in completing the TRIC form. Verbal consent to make electronic recordings of interviews was obtained from each participant prior to commencement. None of the participants objected. Each session lasted about 90 min. Sessions were conducted online due to COVID-19 restrictions. All sessions were electronically recorded, coded for anonymity, and stored securely in a password-protected folder on the lead researchers’ server.

Participants were encouraged to email supporting documents identified during the sessions to the research team (e.g., policies, procedures, data collection forms). These provided additional information sources and were coded to link with the deidentified recordings and completed TRIC forms.

To collect human resource data, the rehabilitation managers in four of the five provinces networked with lead clinicians or managers at each rehabilitation facility to obtain information on the total number of each type of therapist, and the total number of therapists by province. We previously published detailed procedures for collecting and validating information on rehabilitation workforce personnel and human resources [30]. The fifth province required permission from each health facility, which was not logistically possible within the study timeframe, as this province has more than 800 facilities. Therefore, in this province, the research team obtained human resource information from the central human resource database, which was verified by one of the key informants.

### 2.9. Data Validation Procedures

The principal author made notes during each meeting (which were also visible to participants in real time, using the ‘screen-share’ function). The completed sections of the TRIC were emailed to participants after each session for accuracy checking. Participants were also invited to listen to recordings of their sessions for member-checking validation purposes. After data synthesis, the NDoH deputy director, Disability and Rehabilitation, was invited to validate the information collected from provinces.

The human resource data from the TRIC were verified by district managers, profession-specific managers, rehabilitation facility heads, and/or provincial managers. All therapists in four provinces were also invited to attend an online meeting to verify the human resource and other TRIC information (a total of 220 participants attended these sessions and represented three provinces). As an incentive, therapists received continuing professional development (CPD) improvement points for attending the session.

After data collection and preliminary analyses were completed, the key participants were invited to attend a two-day face-to-face meeting. Preliminary findings were presented, and discussions were held to clarify information and validate analysis. Additional information was collected, where necessary, to enhance understanding.

### 2.10. Data Analysis

TRIC data were entered into MS Excel for analysis (see Appendix A).

#### 2.10.1. Quantitative Data

Categorical responses (yes/no/somewhat/no response) were totalled. Where relevant, these were analysed as proportions (each province provided one agreed response) and findings were presented in graphs, tables, or narratively.

#### 2.10.2. Qualitative Data

Qualitative data were gathered from the responses to the open-ended TRIC questions. Qualitative information often explained the quantitative responses. Most qualitative responses included only a few sentences, precluding in-depth analysis. However, we followed as many as possible qualitative data analysis steps [34]. Two researchers (QL and TC) first familiarised themselves with the data. A deductive analytical approach was followed, using the TRIC categories as main themes. Exemplar quotes that reflected participants’ collective responses were extracted for each main theme. Two researchers (QL and TC) independently identified exemplar quotes, discussed their representativeness, and agreed on the final selection.

#### 2.10.3. Human Resources

In each province, we totalled the number of therapists overall, and the total number of therapists by profession. We then calculated the ratio of therapists per 10,000 of the uninsured population in each province (estimated at 80% total provincial population) [35]. In South Africa, the uninsured population depends on the public health system for healthcare, while the insured population utilises private healthcare. Therefore, ratios of workforce and the uninsured population are commonly used in South Africa to describe coverage of the healthcare workforce [32].

## 3. Results

### 3.1. Participant Demographics

There were 17 participants (mean age 41 years; range 30–60 years; two males). Participants had an average of 10 years (range 3–18 years) experience in their current portfolio. Participants comprised 11 managers (district, profession-specific, and provincial), one clinical head, and five lead therapists. Seven participants were occupational therapists, four were physiotherapists, one a speech-language and hearing therapist, two were audiologists, one a medical orthotist and prosthetist, and two were speech therapists/audiologists.

### 3.2. Governance, Regulation, and Leadership

#### 3.2.1. Quantitative Responses

All provinces (100%) had a designated unit or officer for rehabilitation within the provincial ministry structure. Regulatory frameworks for health that apply to rehabilitation existed in all provinces (100%). Four of the five provinces (80%) had clear governance structures in place and used available information and evidence about rehabilitation service availability to inform service planning. However, four provinces (80%) reported that rehabilitation was excluded from health policy and legislation frameworks, even at the provincial level.

#### 3.2.2. Qualitative Responses

Rehabilitation leadership was represented by provincial rehabilitation managers situated in the provincial DoHs, although the Department of Education (DoE) employed rehabilitation professionals as well. All five provincial DoHs had at least one deputy director/rehabilitation manager (two in two provinces). At the facility level, however, rehabilitation governance varied between provinces. In four provinces, tertiary care facilities had clinical rehabilitation managers (overall, or one for each profession). However, this management model was not necessarily replicated at primary level facilities. Since financing and operational service planning occurs at the facility level, rehabilitation leadership at this level is crucial.

Participants indicated the importance of including a range of stakeholders in discussions about service improvements and planning. Stakeholders included, but were not limited to, users of the public system and non-governmental organisations (NGOs). Participants commented that while NGOs often provide feedback on tenders, they can contribute much more to the governance of rehabilitation because they bring different perspectives and insights. Representatives of one province reported the implementation of a Disability Forum to ensure that the voices of persons with disabilities who access the public health system are heard, and that such initiatives could facilitate improved access to quality rehabilitation and healthcare.

Although rehabilitation governance structures were in place in all provinces, participants indicated that rehabilitation is generally not factored into high-level planning or strategies. These concerns were underpinned by the lack of reference to rehabilitation in strategic planning documents. One participant commented that rehabilitation is not viewed at a strategic level and is only considered at operational level.

Participants indicated that although they were invited to contribute to strategic documents, their voices were often not ‘heard’ and consequently their input was overlooked or devalued. One participant commented that rehabilitation was disregarded because of the lack of indicators or data-driven key indicators:
*“Provide input but rehab(ilitation) info(rmation) never included—need indicators” (“Since there are no agreed indicators except for the number of wheelchairs and hearing aids issued.”)*

In addition to the lack of data and key indicators, participants unanimously agreed that policy makers often do not understand what rehabilitation is, what it can offer, and how it should contribute to key performance indicators. One participant from an urban province commented:
*“Stakeholders don’t have insight of what rehab(ilitation) is and what therapists do.”*

One participant noted the lack of collaborative, multi-level leadership within the rehabilitation discipline with a broad range of stakeholders (academics, regulatory bodies, researchers). Although several committees are available in clinical fields, coordination of a more united approach and action will strengthen rehabilitation:
*“… Need to pull all structures together, including academics—new graduates are very discipline specific, despite the promotion of IPE [interprofessional education] at universities.”*

Participants acknowledged that rehabilitation governance is complex because rehabilitation itself is complex. It involves coordination within rehabilitation teams, and additional synchrony with the comprehensive care required by the individual. Participants felt that rehabilitation is affected by broader health system failures, for example, inadequate governance and the lack of follow-through of provincial plans or policies nationally. These broader health system factors fall outside the remit of rehabilitation governance, but can significantly impact on rehabilitation service assessment and quality:
*“Rehabilitation is a victim of governance flaws and failures in the system.”*

#### 3.2.3. Opportunities and Emerging Initiatives

Participants reflected on strategies that could elevate the visibility of rehabilitation at strategic levels. They noted the dynamic nature of health management, where rehabilitation managers should capitalise on opportunities as they arise. Rehabilitation managers at one predominantly rural province reflected on key challenges, such as litigation related to children with cerebral palsy. Since funding for rehabilitation forms a large proportion of these claims, there are opportunities to strengthen rehabilitation options, which could significantly reduce litigious pay-outs. There were also strong suggestions that the inclusion of rehabilitation in national strategic initiatives for which there are ring-fenced funding (e.g., NHI), could improve access to, and visibility of, rehabilitation at the national level.

A participant from one urban province reported on the formation of a new committee to plan and implement integrated care. Rehabilitation is represented on this committee, which marks a positive step towards integrated care pathways, and a move away from traditional specialised care models such as HIV clinics.

### 3.3. Rehabilitation Financing and Resource Optimisation

#### 3.3.1. Quantitative Responses

All provinces (100%) indicated the existence of an allocated budget for rehabilitation and assistive products. Only one province (rural) could provide the total 2021 expenditure for rehabilitation (including human resources, assistive technology, and consumables). In this province, rehabilitation expenses constituted R96 million, about 0.3% of the total provincial health budget. Figure 1 illustrates how this money was expended in this province (where about R28 per person was earmarked for rehabilitation).

#### 3.3.2. Qualitative Responses

Most participants indicated that they lacked ready access to information about rehabilitation expenditure. Their perceptions were that financing for rehabilitation was not standardised or even monitored across facilities, limiting opportunities to examine funding fluctuations or trends over time. Participants indicated that most rehabilitation expenditure—such as the human resources, consumables, and infrastructure—occurs at facility level, but routine accounting and reporting disaggregated by rehabilitation is mostly not in place, therefore information remains limited.

All participants reported the Government (National treasury) as the major financing mechanism for rehabilitation. The sources mentioned included the Road Accident Fund (RAF) and the Workers’ Compensation Assistance (WCA). The Public Finance Management Act (PFMA) was reported as the regulatory body that ensured the efficient and transparent management of all revenue and expenditure provided by the government. Additional financing mechanisms included the NGOs but was not monitored by the PFMA Act. There were typically no out-of-pocket (OOP) costs incurred for rehabilitation provided at government facilities, while one province (20%) mentioned that this was dependent on patients’ income.

The major health financing mechanisms covered all levels of rehabilitation including hospital-based inpatient and outpatient rehabilitation, community-based rehabilitation, and outreach (home visits, school visits, workplace visits, and clinic visits). The training of a few community health workers (CHWs) was covered. No special funding was allocated to rehabilitation that targeted children with developmental delays and disabilities.

### 3.4. Health Information and Research

#### 3.4.1. Quantitative Responses

The quantitative responses indicated notable inefficiencies pertaining to rehabilitation health information. Figure 2 illustrates that the only relatively positive element that relates to the data is the availability of agencies to conduct research.

#### 3.4.2. Qualitative Responses

Participants were asked about the routine collection of data on the types of rehabilitation needs that existed in their provinces. Participants mostly reported awareness of the information generated by the Washington Group Short Set, which had been included in the South African Census. No other data appeared to be routinely collected on rehabilitation needs in South Africa.

One key finding was the lack of high-level indicators for rehabilitation within national or provincial health monitoring frameworks. The only indicator mentioned was the number of wheelchairs and hearing aids issued during specific time periods, which is included in the national health monitoring and reporting systems. Participants unanimously agreed that these were insufficient, since they failed to provide any indication of service coverage or quality, or how rehabilitation contributed to key performance indicators on population health.

Routine data on service delivery (number of patients, type of patients, number of treatment sessions) were captured in all five provinces but no data were recorded on actual need, which is estimated to be significantly higher than current service levels. Although a growing number of research projects are being conducted in this area, they are not always conducted in collaboration with clinical services, and moreover, projects are generally focused on clinical rehabilitation topics.

The key challenges pertaining to rehabilitation data collection are depicted in Figure 3.

Participants commented that they were aware of rehabilitation research projects being conducted in their provinces. However, research funds were limited, and projects were mainly clinically oriented. The key funding bodies mentioned included the state-based South African Medical Research Council (SAMRC) and the National Research Foundation (NRF) and private bursaries available for postgraduate studies. Participants suggested that systems-related rehabilitation research is urgently needed.

#### 3.4.3. Opportunities and Emerging Initiatives

In one province, an online method had been developed to routinely collect human resource information and service delivery data (e.g., number of treatment sessions; patients treated; conditions). The data were verified by on-site visits and regular check-ins with rehabilitation coordinators and managers at a facility level. These data were not integrated into the provincial online systems, although this was a goal going forward.

### 3.5. Rehabilitation Workforce

An assessment of workforce capacity highlighted notable limitations in all rehabilitation professions, as the ratio was <1 across provinces. Table 1 shows that physiotherapy and occupational therapy’s workforce capacity was better than the other rehabilitation professions in South Africa. None of the provinces had information on trends over time, and the participants indicated that the human resource system data were often incorrect. Concerns were expressed regarding increased vacancy rates, especially post-COVID-19, in some provinces.

Regarding workforce competence and experience, participants mentioned self-initiated local strategies to support therapist training and competence which included forums, in-service training, CPD courses funded by the skills development fund, and student supervision. Almost all therapists (97%) had undergraduate university degrees (3% had diplomas), but only 6% had a masters or doctoral degree. The participants highlighted the limited career-advancing pathways for therapists and that obtaining a postgraduate degree does not ensure a higher-level post or better salaries.

The salaries of entry-level therapists were about one-third of the salary of a medical officer, and slightly less than professional nurses. In addition, participants highlighted that there were no incentives in place to attract or retain therapists (except for rural health allowance). There were once-off allowances for uniform and post-graduate studies.

#### Opportunities and Emerging Initiatives

One urban province had initiated special interest groups, chaired by a nominated chair, as acknowledgement of their clinician expertise. This initiative offers affordable and accessible opportunities to maintain or bolster clinical competence.

Participants suggested that better collaborative approaches between academic educators, researchers, clinicians, and health managers would yield benefits in terms of rehabilitation capacity.

### 3.6. Rehabilitation Services

#### 3.6.1. Quantitative Responses

Figure 4 reports the agreed responses from all provinces to the 29 individual, categorical service-related questions of the data collection tool and indicates areas for improvement. A positive response (“yes”) was used if the service-related factor was in place, whilst a “somewhat” response indicated that participants were not convinced that this service aspect was optimal. Only around 30% of the responses were positive, indicating significant opportunities for improvement. The areas that received positive responses from all provinces (100%) included the availability of rehabilitation at tertiary and secondary hospitals, in subacute levels of care, and the provision of rehabilitation for children with developmental delays and disabilities and for people with mental health conditions. Additionally, four of the five provinces (80%) reported regular monitoring of quality for rehabilitation.

#### 3.6.2. Qualitative Responses

All participants provided perceptions that rehabilitation services were not aligned to the needs of the local population, considering the burden of disease and the workforce shortages.

Participants indicated that rehabilitation services were mostly located at tertiary institutions. However, all participants indicated the need to strengthen primary care and community rehabilitation, which hinged on weekly outreach services in three of the provinces (see Figure 3). Specialised rehabilitation services were reported to be available for children with developmental delays and disabilities, people with vision and hearing loss, and people with mental health conditions. Regarding the latter, mental health was considered to be a directorate on its own but received the support of therapists in the psychiatric institutions, psychiatric wards in hospitals, and outreaches. However, rehabilitation that targeted the elderly was non-existent (e.g., fall prevention programmes).

Rehabilitation services delivered in the community were said to be very limited—existing services were provided by rehabilitation care workers supported by district-based therapists as well as primary care therapists. Non-governmental organisations provided training for carers of people with disabilities and mobility training for people with vision and hearing loss.

The need to develop rehabilitation guidelines and integrate rehabilitation into priority programmes and comprehensive care pathways was commonly suggested as a way of improving referral to rehabilitation, and access to services. One rehabilitation manager commented that managers in other services should understand how rehabilitation can contribute to the KPIs of broader health programmes:
*“Make managers understand what rehab is and how it fits into their programme.” Contextual factors influencing services*

Socio-economic factors, such as the high crime rates, unemployment, poverty, domestic violence, lack of education, and substance abuse, have a notable effect on the efficiency of rehabilitation services. People requiring rehabilitation have complex needs, and many are in complex domestic and social situations, where they may also require mental health support and rehabilitation. Participants also noted that students were not trained to deal with this level of complexity. Providing appropriate education for rehabilitation recipients and their families/caregivers was reportedly challenging and required high-level skills. For example, teenage pregnancies were common, and these mothers required special support if their children required rehabilitation.

Several participants suggested that an increasing number of families and caregivers were unable to take care of patients once they were discharged from hospital, due to less-than-adequate socio-economic circumstances. For example, family members were unable to leave much-needed employment to provide care, or non-working family members were not equipped with skills, or were healthy enough, to act as carers. In addition, the influx of foreign patients (mostly migrants from other African countries) meant that many patients with severe disabilities (in long-term rehabilitation facilities) could not be discharged, as their families were not in South Africa. This presented significant ethical and legal dilemmas since the already-limited number of long-term rehabilitation beds could not be made available for new patients.

The high burden on medical and nursing staff limits their ability to educate patients on medical aspects and risk factors related to their condition. Consequently, increased strain is placed on the limited rehabilitation workforce to address this gap; consequently, less time is spent on rehabilitation.

The participants indicated that there is often no opportunity to educate patients and their carers due to early discharge. In addition, education is often limited by the lack of available translators in many settings, and written information may not be available in all languages. Even if written information is available in local languages, low educational levels of many patients and families, coupled with other cultural factors, mean that patients often cannot read, understand, and/or act on the information. Thus, therapists are often tasked with explaining information to patients, families, and other caregivers, which places additional burdens on already-stretched rehabilitation capacity.

#### 3.6.3. Opportunities and Innovations

Participants reported strategies that complemented available rehabilitation services at community level. These strategies included engaging community health workers and including NGOs and community peer support groups. In one province, orientation and mobility practitioners (mid-level health workers) were employed to act as conduits between patients and the healthcare system.

### 3.7. Infrastructure and Medicines

#### 3.7.1. Qualitative Responses

Most rehabilitation facilities at the tertiary level had the necessary infrastructure for rehabilitation, including access to equipment, gym facilities, and kitchen/bathrooms for the assessment of, and training in, activities of daily living. Most patients require primary care and community rehabilitation but the infrastructure at this level of care is reportedly inadequate and/or inappropriate, particularly to conduct group sessions or manage special groups such as children.

The rehabilitation-related medicines that were reported to be mostly available in government services in the country were non-steroidal anti-inflammatory drugs (NSAIDS). Baclofen and corticosteroids were either not available or only available at higher levels of care.

#### 3.7.2. Opportunities and Initiatives

The design of new primary care facilities with purpose-built sections for rehabilitation was mentioned by one manager, who suggested that this indicates progress towards strengthening primary care rehabilitation.

### 3.8. Assistive Technology

#### 3.8.1. Quantitative Responses

All five provinces (100%) were not equipped with an essential list of assistive products, and 80% of these did not have a strategy or plan that included assistive products. However, quality regulations, safety and service standards were somewhat available for the procurement and provision of assistive devices in all the provinces.

#### 3.8.2. Qualitative Responses

Regulatory bodies for assistive devices were in place to oversee the quality and provision of assistive devices. Participants agreed that most of the basic assistive technologies were available and were procured via national tenders (for those devices that were routinely issued). However, ordering devices that were not available by tender was often constrained by complex procurement processes (for instance, dictated by facility budgets). These procurement difficulties resulted in delays in the provision of appropriate, specialised, assistive technologies for people with complex disabilities.

#### 3.8.3. Opportunities and Initiatives

One urban province has implemented provincial tenders to reduce bureaucracy in ordering assistive technologies that are not on the national tender. This initiative has proven to be feasible and efficient in procuring assistive devices in a timely manner.

### 3.9. Emergency Preparedness

#### 3.9.1. Quantitative Responses

Figure 5 provides an overview of the national responses to TRIC tool close-ended questions regarding emergency preparedness. Most of the responses were negative indicating how poorly rehabilitation is involved in emergency preparedness.

#### 3.9.2. Qualitative Responses

The participants reported that rehabilitation was poorly integrated in emergency response plans. One cited reason was that rehabilitation workers were not considered to be frontline workers; hence, focus was on medicine and nursing professions. Participants unanimously agreed that rehabilitation capacity was not mapped and known to the Ministry of Health and Health Emergency Operation Centre or National Emergency Management Agency. No contingency plans nor stockpile for assistive products were in place for the continued provision of rehabilitation in the event of a disaster or pandemic. One province mentioned the availability of prefabricated orthoses only. All provinces had no established referral pathways between rehabilitation services in high-risk areas and those in low-risk areas.

#### 3.9.3. Opportunities and Initiatives

The COVID-19 pandemic raised awareness regarding the need for strengthening emergency preparedness within rehabilitation, e.g., to accommodate an increased need for assistive devices. Telerehabilitation became increasingly popular as a means of continuing rehabilitation services for non-COVID patients.

## 4. Discussion

To our knowledge, this is the first paper to report on a detailed analysis of the public health sector’s rehabilitation capacity in South Africa. The findings provide a broad overview of the current strengths, weaknesses, opportunities, and emerging initiatives/innovations to advocate and enact priority steps for the prioritisation and strengthening of rehabilitation in the health system.

The WHO TRIC tool facilitated the collection of comprehensive data on the country’s rehabilitation situation. This validated tool was previously used in LMICs to assess rehabilitation capacity [36]. Although South Africa is a high-middle income country, it is one of the most unequal countries globally. Therefore, the appropriateness of capacity assessment items may vary across regions with varying complexities of the health system, even within the same province. This study is therefore unique as it reports on the use of the TRIC tool in a complex setting. However, the TRIC tool was useful in guiding a standardised approach to collecting and reporting on rehabilitation capacity from relevant people working at different levels of government and service provision in South Africa.

This research identified that South Africa’s rehabilitation capacity has strengths in terms of leadership and governance. Rehabilitation managers are vocal advocates at provincial and national levels and are appropriately positioned at the governance level to drive strategic initiatives, and implement key policies, such as NHI. However, rehabilitation managers reported their limited influential abilities since policy makers do not regard rehabilitation as integral to healthcare, nor as an essential health service. Many factors contribute to the sub-optimal strategic positioning of rehabilitation within the South African health system. One key barrier could be the lack of appropriate and reliable rehabilitation data to drive data-driven advocacy for informed decision-making by policy makers. Although participants in one urban province reported taking significant steps to increase the collection of reliable data, they have not yet crafted an avenue for data to be utilised at a strategic level. Most participants indicated that routine service-related statistics are collected (such as number of people treated); however, this type of data may be inadequate to drive high-level decision-making [37]. In addition, all participants agreed that the current national indicators for rehabilitation (e.g., number of wheelchairs issued) do not appropriately reflect the essence of rehabilitation services. There are no set standards for bench-marking the recorded rehabilitation statistics (such as HR ratios per 10,000 uninsured population). More importantly, there is not a clear linkage between the current rehabilitation statistics and key NDoH performance indicators. This asynchrony between data sources perhaps explains the perception that rehabilitation is not viewed as a priority, nor a health strategy. More appropriate indicators to showcase the relevance of rehabilitation to societal participants such as employment, education, and the economy, are necessary. The potential contribution of rehabilitation to population health indicators is also advised to advance rehabilitation in South Africa.

South Africa, as do many other countries globally, uses mortality and morbidity as standard indicators of population health. Morbidity and multimorbidity levels are high in South Africa due to the quadruple burden of disease [38]. Our participants reported that policy makers and health planners do not consider rehabilitation at a strategic level, possibly because they do not understand how rehabilitation contributes to population health. Rehabilitation is defined as a restorative strategy to optimise functioning [26] in a wide spectrum of the population with functioning limitations due to injury or disease [26]. However, the association between functioning, morbidity, and health is not often explored or understood by stakeholders. A more explicit explanation of the relationship between functioning (primary outcome of rehabilitation) and morbidity (as an accepted indicator of health) may assist in explaining the contribution of rehabilitation to the health of the population. A recent study [36] conducted in the European region may offer valuable insights in this regard. Nugraha (2020) investigated the relationships between morbidity, functioning, and the subjective perception of health among a large cohort of Europeans. Their results suggested that morbidity and functioning problems both lead to poorer self-rated health, and that morbidity increases the probability of reporting functioning problem(s) [36]. The authors suggested that since their finding can be applied across cultures, it should be considered globally by health policy planners to advance health strategies such as rehabilitation, which is aimed at improving function. Considering that one-fifth of South Africans older than 15 years self-report poor health [38], this should be of concern to the government. Our participants suggested that evidence is needed to assist in explaining the role of rehabilitation to health policy makers. Primary evidence such as that provided by Nugraha et al. (2020) [36] could, if communicated well, be useful in enhancing political awareness, interest, and much needed investment to improve the rehabilitation workforce and infrastructure in South Africa.

South Africa’s envisaged NHI is a major health transformation that presents investment opportunities for strengthening rehabilitation. Our participants suggested that rehabilitation should be better integrated in major health policies and reforms, particularly to facilitate increased investment. Participants reflected on the importance of integrating rehabilitation into major health reforms such as the NHI. It is envisaged that the NHI will necessitate greater data collection which, assuming rehabilitation is included, could be utilised in rehabilitation research. Potentially ring-fenced funding for the NHI could be directed towards impactful research into improving and integrating rehabilitation services, which currently appears to be mostly limited to clinical studies. The roll-out of the NHI will incorporate notable technological transformation, which could support the inclusion of rehabilitation in patient and broader healthcare plans. However, to seize these opportunities, the rehabilitation community should strive to reduce disciplinary boundaries, which were reported by our participants. Unless the therapies are represented by a united voice in the design of major, national health reforms, it could constrain investment and interventions to strengthen rehabilitation in the South African health system.

The implementation of the NHI at an operational level involves the design of standardised care packages along care pathways of priority conditions. Our study shows a startling lack of rehabilitation guidance documents available in South Africa, which may hamper the inclusion of rehabilitation along NHI care pathways. To address this challenge in a timely manner, South Africa could leverage on international initiatives in this area, such as the WHO’s Package of Interventions for Rehabilitation (WHO PIR) [39]. The WHO PIR should be contextualised to South Africa, to reflect its population health needs as well as unique socio-demographic and cultural factors. In a highly unequal society such as South Africa, rehabilitation packages of care should be designed and delivered in a manner that supports access to rehabilitation for the most vulnerable and socio-economically deprived communities and individuals. Shaping access to care in this way will ensure that the government can deliver on its mantra of “leaving no one behind” [40].

Our participants suggested that the capacity of public sector rehabilitation services is inadequate for the local need, which is based on the notable proportion of South Africans who can benefit from rehabilitation services [8]. However, although the need for rehabilitation in South Africa is significant, increasing capacity for rehabilitation in the health system is hampered by a reportedly low demand for rehabilitation services by referring health practitioners and public. Examples of low referral rates for rehabilitation reported in the literature is 4% for children with cerebral palsy [41] and 8% for adults with stroke [42]. Such low referral rates arguably imply that rehabilitation is not perceived as a high-value healthcare strategy by all stakeholders (providers and users). The demand for rehabilitation should thus be improved by multi-level interventions targeting health science students, educators, healthcare providers, and patients and their families or carers, so that those in need receive rehabilitation services. For instance, the inclusion of rehabilitation referral recommendations in the standard, national treatment guidelines to educate healthcare providers is an important initiative that can potentially close the gap between the need and demand for rehabilitation [43]. Solutions mentioned by our participants, such as reliable referral systems (e.g., electronic), improved care coordination, and improvement in the capacity and quality of rehabilitation services, including employing and retaining senior rehabilitation clinicians, can align rehabilitation services to the local needs more optimally. Patient engagement and education on the role and value of rehabilitation in optimising their functioning, health, and quality of life is recommended to improve their desire for accessing rehabilitation services [44].

Advancing rehabilitation nationally and provincially requires efforts from tertiary training institutions. Participants highlighted the need for training to become more focused on team approaches and integrated care models to support the government’s current shift towards integrated care approaches. One urban province mentioned the establishment of an integrated care forum, which can serve as the impetus for educational reform. The educational design of innovative integrated training models that are contextually efficient and feasible for South Africa will ensure that graduates are more open and prepared to work in an integrated manner in local settings. Tertiary institutions play a central role in lobbying for increased funding for health-systems research, which could potentially be focused on strengthening rehabilitation in the health system.

Access to healthcare services is enshrined as a basic human right in the Constitution of South Africa [45]. Access encompasses elements such as availability across geographical locations, affordability, and acceptability [46]. Although we did not specifically assess all aspects of access, we conclude that the lack of rehabilitation capacity at primary care, the limited numbers of specialised rehabilitation facilities, and early discharge for many due to limited bed capacity at tertiary facilities, results in limited access for many South Africans. The poor integration of rehabilitation at the primary care level has a significant impact on access to rehabilitation, as many cannot afford the direct expenses (such as transport) and indirect expenses (such as time off work) to attend rehabilitation at tertiary institutions, where most of the current rehabilitation services are currently available. Therefore, our findings concur with published research that there has been little progress to achieve the goals of the National Rehabilitation Policy [22]. Considering the growth in the need for rehabilitation [8] and the burden of disease, multi-pronged efforts are needed to better integrate and strengthen rehabilitation in the South African health system, especially at primary care.

### 4.1. Limitations

The cross-sectional design of this study does not provide information on trends and cause-and-effect relationships. Data collection was mainly conducted online due to the COVID-19 pandemic and face-to-face meetings may have provided further insights through observation. The study was limited to an analysis of the rehabilitation capacity of the public health sector, whereas the TRIC tool should ideally be used to capture information on rehabilitation across other non-health sectors including labour and education; and non-public sectors including non-governmental organisations and the private sector. The study was limited to five provinces and there may be nuances in the other four provinces that were not captured in this analysis. Though our study was valuable in providing an overview of the situation at a national and provincial level, we cannot generalise our findings to the provinces as we were unable to capture the granular differences in rehabilitation services and capacity at district or local levels. We were unable to obtain data on rehabilitation finances from four provinces and could thus not compare this aspect across provinces. This was because it was not possible for participants to identify expenditure, due to its aggregation in other health expenditure, and the non-standardised accounting approaches used across provinces and facilities. A future study should include facility-level stakeholders that may be able to provide accurate data on the financing of rehabilitation.

### 4.2. Strengths

Multiple strategies were used to validate the data, which contributes to the confidence in the findings. A representative sample of five provinces, which was reflective of a range of factors (socio-economics, geographical location, and training institutions) was used. This assisted in obtaining a comprehensive understanding of the current rehabilitation situation in South Africa.

## 5. Conclusions

Rehabilitation capacity in South Africa is limited compared with population needs. Rehabilitation faces many challenges, but also opportunities for improvement. Access to quality rehabilitation services to meet increasing need is possible with moderate investment. Political will is critical for increasing investment in rehabilitation to redress key shortcomings, such as human resources as well as primary care rehabilitation services. Rehabilitation coalitions within and across sectors are necessary to drive strategic, priority initiatives to position rehabilitation as a valued health strategy in the South African health system. Future focused, locally centred leadership should prevail as the country’s health sector transforms towards a NHI initiative, in which rehabilitation is a core element.

## Figures and Tables

**Figure 1 ijerph-20-03579-f001:**
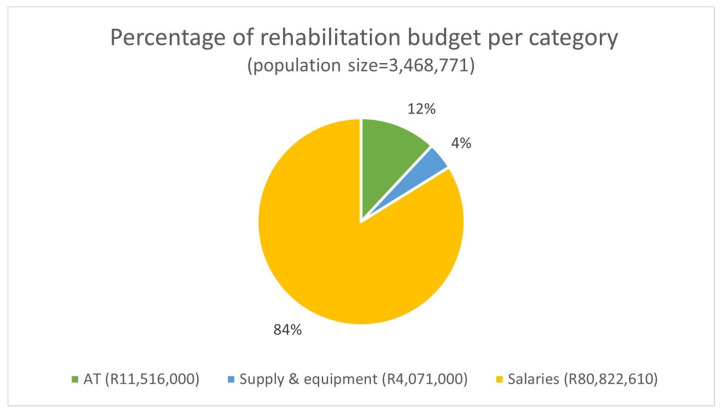
Budget summary of one rural province. Legend: AT—assistive technology.

**Figure 2 ijerph-20-03579-f002:**
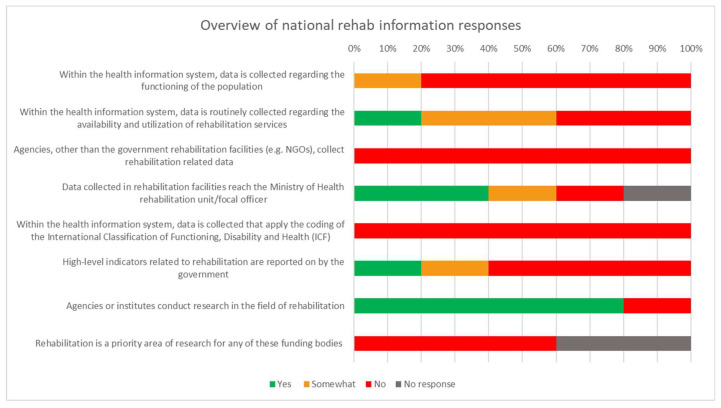
Overview of responses related to national rehabilitation information.

**Figure 3 ijerph-20-03579-f003:**
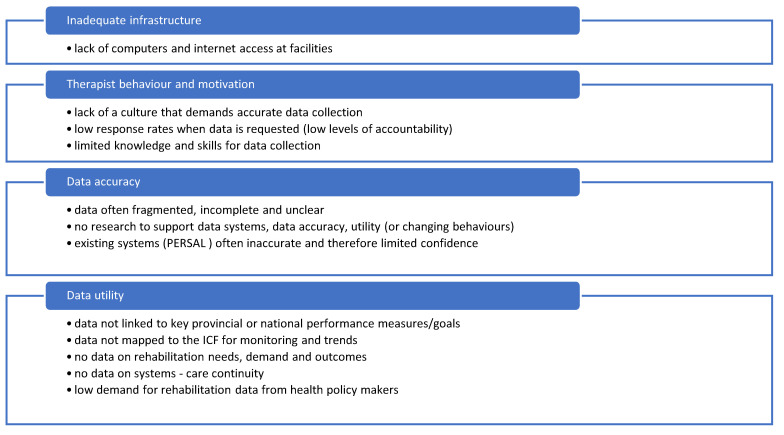
Summary of key findings pertaining to data and information technology.

**Figure 4 ijerph-20-03579-f004:**
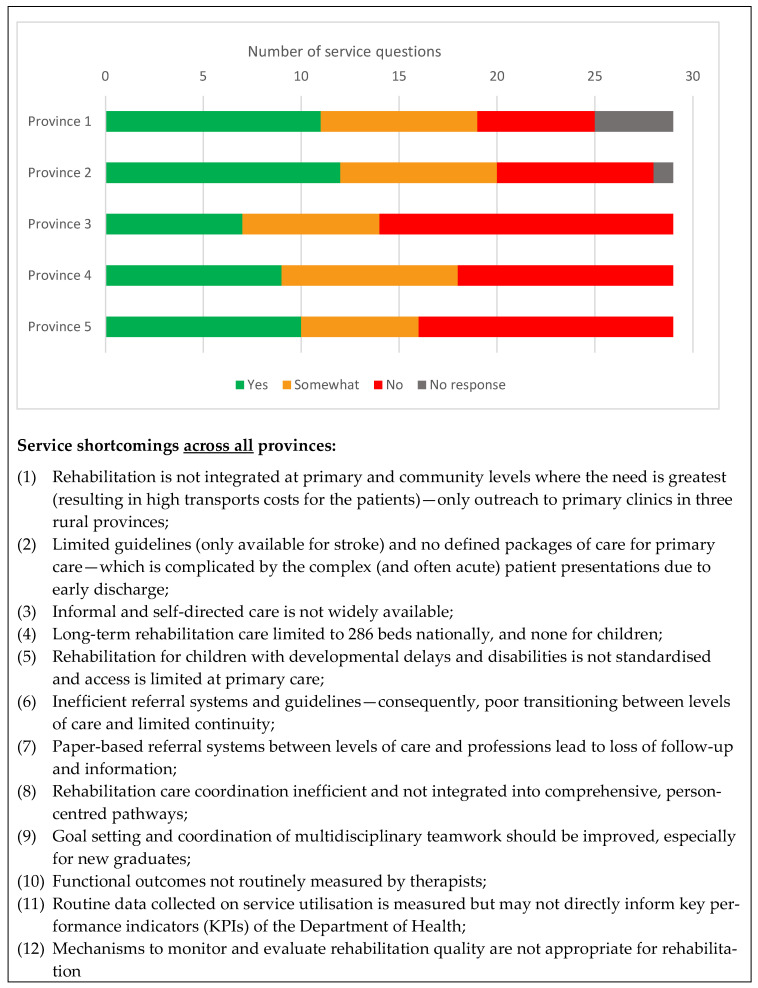
Summary of key findings pertaining to services across provinces.

**Figure 5 ijerph-20-03579-f005:**
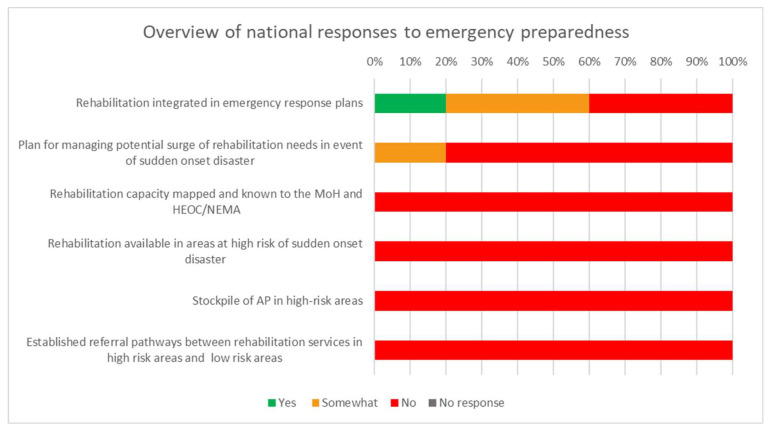
Overview of national responses to TRIC tool questions related to emergency preparedness.

**Table 1 ijerph-20-03579-t001:** Workforce capacity (therapists per 10 000 uninsured population).

Province	Physiotherapy	Occupational Therapy	Speech-Language	Audiologist	Speech and Audiologist	Medical Orthotics and Prosthetics	Total
Rural 1 *(n* = 399)	0.41	0.36	0.07	0.08	0.03	0.06	0.73
Rural 2 (*n* = 454)	0.36	0.47	0.08	0.02	0.07	0.04	1.02
Rural 3 (*n* = 335)	0.37	0.38	0.11	0.09	0.05	0.06	1.03
Urban 1 (*n* = 950)	0.31	0.37	0.03	0.05	0.13	0.03	0.80
Urban 2 (*n* = 383)	0.32	0.35	0.08	0.01	0.01	Not available	0.76

## Data Availability

The data presented in this study are available on request from the corresponding author. The data are not publicly available due to ethical restrictions protecting the anonymity of participants.

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
