# Peer review of "Rehabilitation Capacity in South Africa—A Situational Analysis"

_ijerph, 2023, doi:10.3390/ijerph20043579_

Round 1

Reviewer 1 Report

Summary:

This valuable study provides an excellent overview of many of the key rehabilitation capacity issues facing South Africa. The methodology is sound, and use of the TRIC connects this work with international research and allows for some degree of comparability. Although the data is necessarily generalized at provincial level, it still offers enough specific insight to be helpful. I foresee this paper being of practical use to rehabilitation advocates, managers and health policy makers. 

The paper is well structured and clearly written, with only a few typos and grammatical errors. 

1.    General concept comments: 

I have only three significant conceptual issues with the paper, all of which I think could be dealt with through clearer defense of the authors’ study choices, and acknowledgement of study limitations. 

1.1.         Confining focus to public sector rehabilitation services:

The TRIC aims to capture rehab capacity across sectors (public, private, NGO), and the envisaged NHI would purchase health services from non-state providers as well. Limiting this study to public sector services may have been necessary for capacity/logistical reasons, but should be justified by the authors. 

1.2.         Confining focus to health rehabilitation and specifically professional workforce

 Focusing only on the health sector is another justifiable decision, acknowledged by the authors, but could perhaps be mentioned under limitations – especially for areas of sectoral overlap such as ECD and children with learning disabilities. The conceptualization of rehab in the article is quite medical, with an apparent bias towards inpatient tertiary service provision (possibly due to sampling?). Although the need for PHC-level rehab is acknowledged, community-based rehabilitation is not mentioned at all, despite being espoused by the Framework and Strategy for Disability and Rehabilitation (FSDR). Again, this is a conceptual position the authors should justify, if only briefly. 

Most problematic is the statement in line 139ff that rehabilitation "mainly includes access to therapy professions". Although the position of mid-level rehabilitation workers (MLRW’s) is controversial and inconsistently implemented across the country, profession-specific MLRW’s (e.g. OT technicians) and CBR workers do play a significant role in service provision in many areas. Task-shifting to MLRW’s is also generally agreed to be an essential strategy for a viable and sufficient rehab workforce. 

Line 291: The statement “community workers are not involved in rehab services" is also not generalizable. 

Finally, it would be worth knowing whether mental health rehab was included/represented – often it isn’t. 

1.3.         Acknowledging disparities between areas within provinces

The highly uneven provision of rehab services within provinces and even districts makes generalization at provincial level very difficult. While this study provides a necessary overview of the situation, it is unlikely to capture more granular differences at local level. 

The chart of rehab service question responses on p12 refers particularly here – I am very curious about which items the provincial reps could agree on as “yes”! I suspect these were mainly the questions about tertiary-level services, since these are both the best resourced facilities (as the authors and participants acknowledge), and the fewest in number. If this is the case, the roughly 30% “yes” answers are only good news for the tiny proportion of service users (or rather service needers) who can actually access them. A bit more analysis of this the data might yield more insight. 

Some further points: 

-       Important to acknowledge rural-urban disparities in health service provision (for example in the introduction, see below)

-       How many district managers were included in the sample, and were they from urban or rural districts? (since predominantly rural provinces still have urban centres)

-       Given the data issues, can provincial managers really have a solid grasp of what’s happening in all their districts? Could there be reporting biases in the research data in this respect? For example, I find it very hard to believe that “most basic assistive devices were available” across the five provinces (line 503)!!

1.    Specific comments: 

The following are minor points, not all of which require changes to the paper.

Abstract

1.1.         26-27: “Participants told many illustrative stories...”: This point is not referred to again in the paper – doesn’t belong in the abstract. 

1.2.         30: “Continuity of care across sectors”: What sectors do you mean here? Rather levels of care? 

Introduction

A nice succinct summary.

1.3.         60-64: Yes to racial and gender inequities, worth also mentioning the massive (and growing) economic inequality as well. You could mention geographical inequalities (incl rural/urban) here too. 

1.4.         64ff: Poor management: this also has a strong geographical (and historical) component – poorer districts and rural areas have a far harder time attracting and retaining skilled managers, and poorer populations are often less able to hold local government to account.

1.5.         69-71: “14% of total national healthcare funding” - reference? Also important to clarify that the funding in question is not solely government funding – it includes health expenditure across the board, including health insurance payments and OOPE (as an aside, if our private healthcare sector were less corrupt and anti-competitive, that total expenditure would be lower and the % spent on public sector would rise as a proportion of total). 

1.6.         81: Statement on NHI "which provides a health safety-net"- rather say "which is intended to provide a safety net"

1.7.         102: "national and provincial levels": see point 3 above about district representation in the sample

Methodology

1.8.         Consider commenting on the parts of TRIC you left out (e.g. emergency preparedness, questions on non-state rehab services). Also any data collected that wasn’t reported in this paper?

1.9.         Sampling: see point 1.3. above re district managers. Also, was mental healthcare/psychosocial rehab represented here? 

1.10.   252: "Continuous professional practice (CPD) improvement points" – this should be “continuing professional development (CPD) points”

Results

1.11.      Section 3.1. – Check tally of professions? Adds up to 18 not 17

1.12.      326: "Lack of collaborative, multi-level leadership" - not sure what this refers to - leadership of what? 

1.13.      336: Rehab is affected by broader health systems failures – YES

1.14.      Section 3.3. on rehab financing: - is this all? TRIC includes questions about financing mechanisms and sources, OOPE etc. Not all this data is available for SA, but would be worth mentioning RAF and COID in particular as state financing mechanisms (although purchasing private sector services). 

1.15.      Section 3.4. Health information and research:  Captures the data issues really nicely, especially table on p9

1.16.      Section 3.5. Rehab workforce: These figures are hard to interpret without reference ratios - what targets are you comparing them to? (obviously the absence of national standards for rehab is a problem in itself).

1.17.      Section 3.6. Rehab services:

1.17.1. 433: 29 health service questions? I could only see 27 in TRIC

1.17.2. Numbered points (p11) – good summary. Point 1: what is "(high transport costs)" doing in the middle of this sentence? 

1.17.3. Point 6 - "many patients for care transition between levels and continuity": what does this mean? 

1.17.4. Point 10 - functioning outcomes or functional? 

1.17.5. First para on p12: This is incorrect – rehab professionals are first-line practitioners in both private and public sector (written into scope of practice), and referral is not required for rehab access in public sector (at least not uniformly or legally!) 

1.17.6. 454ff: Contextual factors influencing service delivery – interesting section! 

Complexity at community level speaks to need for community-based rehab and intersectoral approach.

Skills demands of context point to vital importance of employing and retaining senior clinicians (cf the authors’ previous work on rehab workforce and predominance of very junior staff – highly problematic!)

1.17.7. 489: What are “occupational mobility practitioners”?

1.17.8. 492-494: Not clear what these “collaborative approaches” are meant to address (is this not more related to workforce section?)

1.18.      Section 3.7: Infrastructure and assistive devices

1.18.1.1.      498: “Since patients were discharged early, most patients thus relied on primary care rehabilitation” – This assumes that the rehab process belongs at inpatient/ tertiary level, and PHC is a fall-back, whereas (a) the majority of service needers will never access tertiary and (b) much of the rehab process must happen where people return to their lives and cope with disability over the long term. This means PHC, not as a making-do, but as first choice. 

1.18.1.2.      503: “Participants agreed that most of the basic assistive technologies were available” – really?!! I find this very hard to believe. They may be theoretically available (through the tender system), but actual provision is frequently scuppered by lack of budget and problematic procurement systems (among other things).

1.18.1.3.      Note: Tender items are generally covered by facility or sometimes regional budgets (depending on the province/area). It is not only non-tender items that face the difficulties in line 506. 

Discussion

1.19.      Delete first para - author instructions

1.20.      521: “…this is the first paper to report on a detailed analysis of the public health sector’s rehabilitation capacity in South Africa.” Agreed, although a study was done by Health Systems Trust in 2016/2017 (on behalf of NDoH), looking at rehab capacity in KZN as a case study, ahead of FSDR implementation. Interestingly, this report was launched, but to my knowledge never made publicly available. 

1.21.      530ff: “Rehabilitation managers are… well placed to drive strategic initiatives and implement key policies” - I disagree. Despite their motivation, rehabilitation managers are disempowered by multiple factors at provincial and national level (not least the issues named in this paper, such as lack of integration into health system structures and decision-makers lack of interest in rehab). Policy implementation is impossible without political commitment to operational plans and budgets. 

1.22.      544: No link between data and key performance indicators - yes, also there are no agreed standards against which to evaluate rehab service stats (e.g. HR ratios per 10000 uninsured population)

1.23.      594ff: Rehab services are not aligned to local need: This needs a bit more unpacking. Is the lack of alignment about service mix, delivery platform or capacity? The link to low demand is not clear. 

1.24.      601ff: The low demand issue is a big one, but educating healthcare providers and patients is a limited solution. Demand only grows reliably when services are visible, accessible and trusted – for this they first have to exist! 

1.25.      605: “…guidelines to education health providers…”: check sentence? 

1.26.      622-631: Accessibility of services – yes!! Note however that even PHC rehab access is not cost-free, especially in rural areas. 

Limitations

1.27.      Good section, but see also points 1-3 on additional issues to address.

1.28.      Would be useful to include comment on where further research should focus as part of the STARS process in South Africa. 

Conclusion

Good summary. 

Thank you for the opportunity to review this paper. I commend the authors on an excellent and useful piece of work. 

Reviewer 2 Report

I’d would like to thank the editor and authors for the opportunity to review this timely manuscript on the rehabilitation capacity in South Africa. I've provided suggestions for consideration below. Main concern to me is the disjunction between the methods (TRIC form) and results section; resultantly, the discussion is difficult to follow and does not always seem to flow from the findings presented.

Introduction

-       I would possibly consider making the first paragraph on “rehabilitation” the third paragraph, starting with South Africa’s overarching health challenges before zooming into the impact of those challenges on the need / challenges for rehabilitation within the South African context, and how that alignes with challenges in globally (/LMIC).

-       Page 1, line 39; The starting point for this introduction seems that a) rehabilitation is essential, and b) rehabilitation is effective. Yet, as you may be well aware… evidence for this is largely derived from high-income settings. While, I’d argue that rehabilitation may be more essential in the wake of inadequate acute / chronic medical management, there seems little evidence to support the notion that indeed, rehabilitation is essential and effective in a public sector, South African context. Is there compelling, quality, evidence from South Africa, that indeed, these (examples of or other) rehabilitation outcomes can be achieved, cost-effectively, within a public sector context? Particularly, when chronic diseases like stroke are considered, many of the potential “benefits” of rehabilitation in functional outcomes and disability, may be offset by the “less-than-desirable” social determinants of health?

-        Page 2, line 73; It would be good to also reflect on the new NCD strategy (2022 – 2027), and the role of rehabilitation in that strategy.

-       I think the description / definition of “rehabilitation” provided, is too limited to understand “what capacity” is truly going to be assessed. Arguably, this could be professional services (e.g. physiotherapy) within the public sector. However, one could argue there are also informal or less conventional “resources” for rehabilitation including services provided by non-governmental organisations, self-management programs, community-led programs, programs like Western Cape on Wellness, care-giver led initiatives etc. I understand that, from a analysis point of view these may be much more difficult to map, yet it would be good to increase transparency on the scope of what is being analysed and / or touch on the more informal / less tangible “interventions” that may address function, activity, participation and/or quality of life.

Methods

-       Section 2.4; these “areas” do not align with the areas described in the TRIC (https://apps.who.int/iris/bitstream/handle/10665/330956/9789241516013-eng.pdf?sequence=1&isAllowed=y). It is unclear why areas like medication, or emergency preparedness are not included in this analysis for the South Africa context?

-       Section 2.6; A stronger description of what constitutes “rehabilitation” and what not could also help clarify why programs like dietetics / nutrition or biokinetics are not included here?

-       Section 2.5; is it really a “survey”? If I read correctly, a TRIC form was completed during an online “interview”-type sessions with sometime multiple participants within the same interview? 

-       Section 2.7; What makes an informant “key”? The number of participants in the study seems relatively small given the sampling strategy? Please clarify.

Results

-       In general, I struggle with the results section as I feel there is a disconnection with the TRIC form on the one hand, and the results presented on the other hand. The appendix A fails to clarify this. Quantitative data seems mostly missing, and a strong emphasis is placed on qualitative data? The qualitative data presented doesn’t align with the open-text items in the TRIC form, and rather suggest “meta-data’ that originates from the interview process when completing the TRIC forms? All this data is valuable. Conversely, sometimes there is reference to quantitative data that I find difficult to detangle from the form (e.g. 29 service-related question, yet only 27 items can be found in the form; none of which seem to have the answer options as presented in the figure 3)

Would it be a suggestion that, for each TRIC area, a quantitative summary is provided first, followed by qualitative findings from the open-text responses, followed by “meta-data” like “financing information was reported not available”, opportunities / challenges?

-       I would strongly encourage to co-submit (in the paper, or as supplement) a fully completed TRIC form or dataset (while preserving anonymity of the stakeholders interviewed, however, with the literature and resources used); an amalgamation of all the findings, to serve as a reference document and benchmark for future initiatives; including section 1.1 to 1.3. Where possible stratified by included province. 

Hence in short, either the methods section needs major revision, to better understand the results section; or the results section needs major revisions to connect the TRIC tool to the findings presented.

Discussion

-       Would it be worthwhile to organise the discussion around the same thematic areas as the TRIC form / results section? In line with my struggles with the results section, I find it difficult to gage how one reached the conclusions/ discussion points presented.

-       Please add subheadings to improve readability

-       What is the role of local evidence on the cost-benefits for rehabilitation in driving the South African rehabilitation agenda?

-       Do you agree that the new NCD agenda shows signs of hope, for acknowledging the importance of rehabilitation as an essential health service?

Minor comments

-       Page 1, line 17; It took me a couple of times reading to grasp the meaning of “peripherally” in this sentence; please consider revising.

-       Page 1, line 27; outcomes “may” include?

-       Page 2, line 54; is there a reason for using NCCDs as an abbreviation, and not the more common NCDs?

-       Figure 3 heading not aligned with figure.

-       Page 13, line 517-519; Please remove, this is text to guide authors

-       Data availability statement is missing

Reviewer 3 Report

 Rehabilitation in South Africa (SA) operates peripherally to major health services and reforms, despite the increasing rehabilitation need. However, this paper is only a descriptive analysis, without in-depth analysis and lack of stronger data support. It feels more like a report than a paper, so it is not recommended to be published.

Round 2

Reviewer 2 Report

Compliments for the work done and important paper for strengthening the rehabilitation space within South Africa (and possibly beyond). All comments have been addressed.

Reviewer 3 Report

  • I recommend that the paper be published in its present form